# Effects of Metolazone Administration on Congestion, Diuretic Response and Renal Function in Patients with Advanced Heart Failure

**DOI:** 10.3390/jcm10184207

**Published:** 2021-09-17

**Authors:** Alberto Palazzuoli, Gaetano Ruocco, Paolo Severino, Luigi Gennari, Filippo Pirrotta, Andrea Stefanini, Francesco Tramonte, Mauro Feola, Massimo Mancone, Francesco Fedele

**Affiliations:** 1Cardiovascular Diseases Unit, Department of Medical Sciences, Le Scotte Hospital, 53100 Siena, Italy; gennari@unisi.it (L.G.); pirrottaf90@gmail.com (F.P.); astefanini94@gmail.com (A.S.); molex829@gmail.com (F.T.); 2Cardiology Unit, Riuniti of Valdichiana Hospital, USL SUD-EST Toscana, Montepulciano, 53045 Siena, Italy; gmruocco@virgilio.it; 3Department of Clinical, Internal Anesthesiology and Cardiovascular Sciences, University La Sapienza, 00185 Rome, Italy; paolo.severino@uniroma1.it (P.S.); massimo.mancone@uniroma1.it (M.M.); francesco.fedele@uniroma1.it (F.F.); 4Cardiology Unit, Regina Montis Regalis Hospital, 12084 Mondovì, Italy; mauro.feola@aslcn1.it

**Keywords:** heart failure, diuretics, congestion, management

## Abstract

Background: Advanced heart failure (HF) is a condition often requiring elevated doses of loop diuretics. Therefore, these patients often experience poor diuretic response. Both conditions have a detrimental impact on prognosis and hospitalization. Aims: This retrospective, multicenter study evaluates the effect of the addition of oral metolazone on diuretic response (DR), clinical congestion, NTproBNP values, and renal function over hospitalization phase. Follow-up analysis for a 6-month follow-up period was performed. Methods: We enrolled 132 patients with acute decompensated heart failure (ADHF) in advanced NYHA class with reduced ejection fraction (EF < 40%) taking a mean furosemide amount of 250 ± 120 mg/day. Sixty-five patients received traditional loop diuretic treatment plus metolazone (Group M). The mean dose ranged from 7.5 to 15 mg for one week. Sixty-seven patients continued the furosemide (Group F). Congestion score was evaluated according to the ESC recommendations. DR was assessed by the formula diuresis/40 mg of furosemide. Results: Patients in Group M and patients in Group F showed a similar prevalence of baseline clinical congestion (3.1 ± 0.7 in Group F vs. 3 ± 0.8 in Group M) and chronic kidney disease (CKD) (51% in Group M vs. 57% in Group F; *p* = 0.38). Patients in Group M experienced a better congestion score at discharge compared to patients in Group F (C score: 1 ± 1 in Group M vs. 3 ± 1 in Group F *p* > 0.05). Clinical congestion resolution was also associated with weight reduction (−6 ± 2 in Group M vs. −3 ± 1 kg in Group F, *p* < 0.05). Better DR response was observed in Group M compared to F (940 ± 149 mL/40 mgFUROSEMIDE/die vs. 541 ± 314 mL/40 mgFUROSEMIDE/die; *p* < 0.01), whereas median ΔNTproBNP remained similar between the two groups (−4819 ± 8718 in Group M vs. −3954 ± 5560 pg/mL in Group F NS). These data were associated with better daily diuresis during hospitalization in Group M (2820 ± 900 vs. 2050 ± 1120 mL *p* < 0.05). No differences were found in terms of WRF development and electrolyte unbalance at discharge, although Group M had a significant saline solution administration during hospitalization. Follow-up analysis did not differ between the group but a reduced trend for recurrent hospitalization was observed in the M group (26% vs. 38%). Conclusions: Metolazone administration could be helpful in patients taking an elevated loop diuretics dose. Use of thiazide therapy is associated with better decongestion and DR. Current findings could suggest positive insights due to the reduced amount of loop diuretics in patients with advanced HF.

## 1. Introduction

Congestion is the main cause of hospital admission in patients affected by heart failure (HF), and its resolution with euvolemic status achievement is one of the primary goals of acute treatment [1,2]. Intravenous loop diuretic administration remains the therapeutic cornerstone for acute congestion treatment, although its dosage and timing period administration is not yet universally accepted [3,4]. A dose-escalation strategy and precise protocol for loop diuretic amount is lacking, and administration is often based on urine output and symptoms relief. Unfortunately, these two approaches are not validated enough and poorly related to prognosis. Thus, loop diuretic prescription is generally based on the individual practice of physicians and local hospital habits rather than on standardized criteria [5]. Notably, a position paper of ESC HFA highlighted the importance of proper loop diuretic administration by a correct evaluation of congestion status, assessment of diuretic response (DR), and pharmacologic diuretic strategy [6].

It is generally thought that elevated loop diuretic dose is related to poor prognosis, and few studies have demonstrated a close relationship between diuretic amount and mortality risk [7,8]. This is probably consistent with progressive resistance in patients with chronic use of loop diuretics due to a nephron breakdown and cell hypertrophy of Henle and distal tubule tracts [9]. An emerging finding on which recent research has focused attention in recent years is related to drug responsiveness determined by the relationship between loop diuretic amount and urine production. This relationship was classified as DR, and it was defined as the medication capacity to induce an adequate net urine output and natriuresis to reach euvolemic status [10,11]. Recent studies have demonstrated that poor DR is associated with reduced survival independently of its modality assessment and that DR is often associated with more advanced HF severity and tubular resistance [12,13]. Patients experiencing low DR are characterized by previous HF hospitalization more advanced NYHA class, impaired renal function, and higher comorbidities burden. In this setting, the metolazone addition to the traditional loop diuretic therapy may facilitate both DR and decongestion [14]. Accordingly, we retrospectively selected patients hospitalized for advanced HF (AHF) and reduced DR by three different Italian hospitals (Siena, Rome, Mondovi), and we analyzed the effect of oral metolazone administration compared to the standard therapy on congestion, daily diuresis, and oral loop diuretic dose at discharge.

## 2. Methods

This is a retrospective, multicenter case–control analysis of three Italian hospitals (Siena, Rome La Sapienza, and Mondovi), including consecutive patients with AHF randomized to high-dose intravenous furosemide infusion or intravenous furosemide infusion plus oral metolazone (7.5–15 mg per week) (Figure 1). Patient treatment was at the discretion of physicians, and subsequent analysis comparing the two arms was blinded. All patients were hospitalized and enrolled from April 2018 to September 2020.

Patients taking other thiazide-type diuretics such as hydrochlorothiazide or chlorthalidone, nesiritide, or arginine vasopressin antagonists were previously excluded. Additionally, patients taking inotropic agents such as dopamine, dobutamine, noradrenaline, or levosimendan were also excluded. Therefore, subjects with end-stage renal disease or the need for renal replacement therapy (dialysis or ultrafiltration), isolated diastolic dysfunction, or recent myocardial infarction within thirty days were excluded.

### 2.1. Patient Screening and Evaluation

All patterns were defined according to the last ESC HFA criteria encountering at least two indicators among the clinical functional imaging and historical variables [15]. All patients were in advanced III or IV NYHA class, requiring a high oral dose of furosemide above 100 mg daily. Additionally, all patients experienced a moderate to severe systolic dysfunction with ejection fraction (EF) below 40%. Therefore, in all subjects, DR was calculated by the formula urine output/40 mg of furosemide. According to a previous study, we included in the analysis only patients with low DR (480 mL per 40 mg of furosemide) [10]. In all patients, mean daily diuresis during hospitalization, weight loss, NTproBNP changes, congestion status, DR, and renal function. Weight, NTproBNP, and congestion changes were measured at admission and before discharge [16]. Grading congestion by the assessment of the following clinical signs: pulmonary rales, third heart sound, jugular venous distention, peripheral edema, hepatomegaly, and dyspnea at rest or orthopnea for a total of a maximum of 6 points based on ESC statement [17]. Persistence of congestion was defined as the persistence of at least 2 signs of congestion at discharge if the patients did not achieve the complete resolution of clinical signs at discharge.

### 2.2. Laboratory Assessment

Chronic kidney disease (CKD) was defined as an estimated glomerular filtration rate (eGFR) < 60 mL/min/1.73 m^2^ at baseline. Worsening renal function (WRF) was defined as a serum creatinine increase of ≥0.3 mg/dL or eGFR decrease of ≥20% at any time from admission to discharge [18]. eGFR was calculated using the Modification of Diet in Renal Disease (MDRD) equation. A NTproBNP cut-off >2000 pg/mL at baseline associated with typical signs, and symptoms of AHF and positive radiological chest X-ray were considered criteria for patient inclusion. In all patients, electrolyte, blood urea, and oxygen saturation were monitored during hospitalization

### 2.3. Follow-Up

Clinical outcome was evaluated in terms of death or recurrent heart failure hospitalization over the 6-month follow-up period. There was a scheduled outpatient visit or phone contact at 30, 60, 90, and 180 days after discharge. Combined endpoint was considered death and hospital admission for HF recurrency due to pump failure, volume overload, acute coronary syndrome complicated by HF, ventricular arrhythmia associated with left ventricular dysfunction, or HF related to WRF.

### 2.4. End Points

Primary objectives were the comparison between two groups of DR and congestion score clinically assessed at discharge. Therefore, differences between admission and discharge NTproBNP values were compared. Secondary endpoints included the evaluation of renal function and electrolyte balance after treatment. Adverse event analysis during the 6-month follow-up period was also assessed.

### 2.5. Statistical Analysis

Categorical data are presented as numbers and percentages and were analyzed with the chi-square test. Normally distributed continuous data are presented as mean ± SD. Differences in baseline characteristics for continuous variables were evaluated using appropriate procedures such as the Student’s *t*-test. Differences in primary outcome between Group M and Group F were tested with ANOVA and ANCOVA adjusting for electrolytes, creatinine NTproBNP, body weight, and hypertonic solutions. Different multivariable Cox proportional hazard regression models were used to investigate the relationship between furosemide/metolazone groups and outcome. Multivariable models were adjusted for clinical variables of interest (age, gender, hypertension, diabetes, dyslipidemia, coronary artery disease, and atrial fibrillation) chosen prospectively a priori. Kaplan–Meier survival curve was employed to show the relation between treatment groups and outcome. We considered statistically significant results associated with a *p* ≤ 0.05. We used the SPSS software (version 20.0) for all analyses.

## 3. Results

We studied 132 patients, 67 receiving a high dosage of furosemide (Group F) and 65 receiving furosemide plus metolazone (Group M) with advanced HF. No differences were found in terms of mean age (69 ± 16 vs. 71 ± 21, *p* > 0.05), heart failure etiology (dilated cardiomyopathy: 51% vs. 51%; previous coronary artery disease: 39% vs. 40%; valvular disease: 10% vs. 9%; atrial fibrillation: 32% vs. 33%), respectively, between Group F and Group M. Precisely 72% of patients in Group F vs. 73% of patients in Group M presented IV NYHA class. Similar prevalence of baseline clinical congestion (3 ± 1 in Group F vs. 4 ± 2 in Group M), chronic kidney disease (CKD) (51% in Group M vs. 57% in Group F), and admission creatinine value was observed (1.96 ± 1.05 mg/dL in Group F vs. 1.67 ± 1.2 mg/dL in Group M, *p* > 0.05). Median NTtproBNP was similarly increased in both groups (Group M, 12,177 ± 6283; Group F, 10,316 ± 8815 pg/mL). Loop diuretic amount and other cardiovascular drugs (ACE-inhibitors, angiotensin receptor blockers (ARB), beta-blockers, and mineralocorticoid antagonists (MRAs)) were similarly administered between groups. All these variables are reported in Table 1.

Diuretic Response and Congestion differences- Better diuretic response was observed in Group M compared to Group F (940 ± 149 mL/40 [median 944 IQR 550–1080] mgFUROSEMIDE/die vs. 541 ± 314 mL/40 [median 540 IQR 940–240] mgFUROSEMIDE/die; *p* < 0.001) (Figure 2A). This finding was associated with a better congestion score at discharge in Group M compared to patients in Group F (C score: 1 ± 1 [median 0.8 IQR 0.5–1.2]).

In Group M vs. 2.4 ± 1 [median 2 IQR 1.0–3.2] in Group F) (Figure 2B), whereas median NTpro-BNP and ΔNTpro BNP, calculated as the differences between admission and discharge values, did not show significant differences between the two groups (−4819 ± 8718 [median 4780 IQR 2880–12,700] pg/mL in Group M vs. −3954 ± 5560 [median 3900 IQR 2120–7500] pg/mL in Group F; ΔNTproBNP-26.6% ± 27 in Group M vs. −25.1% ± 25 in Group F NS) (Table 2).

Diuresis, body weight and renal function- Patients taking metolazone experienced better daily diuresis during hospitalization (2820 ± 900 [median 2450 IQR 1900–3000] mL vs. 2050 ± 1120 [median 2080 IQR 1650–2800] mL, *p* < 0.05). Current findings were associated with significant weight reduction in Group M (−6 ± 2.3 vs. −3 ± 1.5 kg, *p* < 0.05).

No differences were found in terms of WRF development, electrolyte imbalance at discharge (Na+ discharge: 138.6 ± 4.5 [median 138 IQR 135–141] mEq/L in Group M vs. 137.8 ± 4.3 [median 138 IQR 135–141] mEq/L in Group F, *p* = 0.3; K+ discharge: 3.87 ± 0.55 [median 4.1 IQR 3.6–4.2] mEq/L in Group M vs. 4.05 ± 0.67 55 [median 3.9 IQR 3.7–4.4] mEq/L in Group F, *p* = 0.09) and creatinine at discharge (1.72 ± 0.78 [median 1.5 IQR 1.2–2.1] mg/dL in Group M vs. 1.69 ± 0.62 [median 1.5 IQR 1.3–2.2] mg/dL in Group F, *p* = 0.79). Patients in Group M were also associated with lower urea (66.4 ± 17.7 [median 67 IQR 52–75] mg/dL in Group M vs. 82 ± 40.5 [median 76 IQR 58–95] mg/dL in Group F, *p* = 0.005). A lower loop diuretic amount at discharge was observed in Group M compared to Group M (175 ± 104.8 [median 150 IQR 100–250] [mg vs. 223.9 ± 121.7 median 175 IQR 125–275] mg *p* < 0.01). However, a higher rate of saline/hypertonic solution administration in Group M was observed (33% vs. 12%; *p* = 0.03) (Table 3).

Follow-up analysis of combined endpoint of mortality and rehospitalization did not differ between the two groups, but a reduced trend for recurrent hospitalization was observed in Group M (26% vs. 38% *p* = 0.34) (Figure 3).

## 4. Discussion

Loop diuretics remains the first therapeutic option to alleviate congestion and symptoms related to hypervolemic status; however, in some cases, despite a dose-escalation approach, they are unable to achieve a complete fluid overload solution [5,19]. Nephron blockade combined with metolazone could overcome this concern by reinforcing the natriuretic efficacy of loop diuretics and by the direct action on Cl/Na exchange at the distal tubule level. Contrastingly, current actions may lead to deleterious effects on electrolyte balance with a severe reduction in plasmatic chloride and sodium levels that are both related to adverse outcomes [20,21].

Our findings demonstrated that in patients with advanced HF taking a high furosemide dose, metolazone associated with standard therapy provided better DR, together with an improvement in congestion score. The increased DR over hospitalization is also associated with body weight reduction. Although our analysis did not show a significant difference in terms of adverse events, the latter findings could have important insights for treatment of advanced HF patients. Indeed, both low DR and natriuresis are associated with increased risk. Although DR was not universally measured, every calculation revealed that it is an excellent modality for risk prediction, and it captures additional prognostic information beyond fluid output and diuretic administration [10,11,12,13]. Otherwise, metolazone treatment is associated with an increased need for electrolyte solution administration in order to avoid Na and K depletion during treatment. The role of hypertonic saline addition in HF is still debated. Preliminary reports have revealed a worse outcome with the use of saline solution, but recent advances have encouraged its application in patients with refractory disease, suggesting positive effects on weight loss and electrolyte balance [22,23,24]. Our results seem to be confirmatory, and the loss of salt caused by the double nephron blockage may be safely corrected by hypertonic infusion without adverse event increase [25]. The last relevant result is the reduction in oral loop diuretic dose administration before discharge. Indeed, several studies have demonstrated a direct correlation between dose diuretic amount and poor outcome. This is due to increased neurohormonal axis and excessive vasoconstriction, more advanced hemodynamic dysfunction, and refractory congestion [20,26,27]. Thus, some authors have assumed that diuretic dose is simply the mirror of an advanced HF stage requiring an increased amount [28]. The relationship existing between the diuretic dose, diuretic response, and outcome remains elusive. Whether an elevated diuretic amount is necessary to achieve an optimal decongestion or if it is simply a reflection of advanced HF is still questioned. In this respect, Hanberg et al., in a post-hoc analysis of DOSE, demonstrated that a more aggressive loop diuretic strategy is beneficial, but when adjusted for diuretic dose and degree of decongestion, the potential was not confirmed [29]. Due to the lack of comparative trials and limited literature, an escalation loop diuretic dose and individualized therapy remain the best option to restore diuretic efficacy and achieve clinical decongestion. Since thiazide diuretics inhibit sodium resorption in the distal tubules, these drugs should be the preferred agent to overcome nephron breakdown in advanced HF patients with refractory congestion [30]. Although congestion in our study was clinically evaluated, and this method may be affected by dissimilar assessment, the score applied agrees with the most recognized criteria, and it accounts for the traditional clinical signs usually assessed [17]. Therefore, the congestion score improvement found in Group M is associated with body weight reduction and increased daily diuresis, representing confirmation of good decongestion [31]. Despite the beneficial effect of metolazone on fluid retention resolution, NTproBNP is not significantly reduced with this treatment. This may depend on the wide range of values, concomitant renal dysfunction in most patients, and other metabolic and systemic disorders influencing plasma levels. Although clinical evaluation is subjective, our protocol is much more precise with respect to other studies applying a more simplistic algorithm based only on weight loss, BNP changes, dyspnea reduction, or peripheral edema relief [32,33]. Unfortunately, the current clinical assessment evaluating peripheral and central signs of congestion demonstrated modest accuracy. Current discrepancies lead to scarce homogeneity, although most studies have confirmed the relationship existing between residual fluid overload and outcome [33,34]. A more detailed and standardized clinical evaluation of fluid retention in different districts and understanding of the related physiopathological mechanisms has become a priority [35,36]. Accordingly, Harjola et al. proposed a score based on congestion-related organ injury by an integrated ultrasonographic assessment evaluating cardiac, lung, and abdominal districts [37]. However, the evaluation of organ deterioration is not limited to physical examination, and it accounts for laboratory markers, echocardiography, and lung and abdominal ultrasound. Nevertheless, in our study, the significant congestion decrease revealed with metolazone was supported by better weight loss and urine output. Previous studies using a combination of loop and thiazide diuretics have shown contrasting results. In an observational cohort of 1048 patients taking metolazone, the use of agents was associated with an increased risk of electrolyte imbalance, worsening renal function, and mortality [38]. Interestingly, in a head-to-head comparison between metolazone and chlorothiazide added to loop diuretics, both agents increase urine output without effects on renal function [39]. The last trial comparing metolazone with tolvaptan and chlorothiazide in 60 acute HF patients resulted in excellent weight loss and diuresis in all three treatments with additive natriuresis evidenced in the metolazone group [40]. Overall, our findings, together with the latest published reports, seem to confirm some advantages in congestion resolution DR and weight loss by diuretics combination therapy in patients with more severe HF. The potential side effect related to excessive loss in electrolytes may be safely balanced with contemporary use of saline solution without evidence of increased risk.

### Limitations

Our data are limited from the retrospective nature of the current analysis. Therefore, the total number of patients enrolled is relatively low, and it does not consent to achieve definitive results. However, to the best of our knowledge, this is the larger study analyzing the effects of metolazone in selected patients with advanced HF and reduced DR. In our protocol, we did not measure the diuretic content of Na and Cl that should help in distinguishing patients with tubular resistance and poor DR. It is our intention to select patients by current measurement for the next study. The observational follow-up period is restricted to the first 6-month period and a longer follow-up could consent to obtain more consistent results regarding the drug effects on outcome. Moreover, the follow-up analysis may be influenced by the subsequent administration of metolazone in Group M, which could be a potential bias, although no differences were found during the post discharge observational period. Of note, some patients could modify diuretic treatment during post discharge period according to congestion presentation. Unfortunately, we only have the diuretic dose at discharge but not drug-modifying disease data. More hypertonic saline solution administration in the metolazone arm suggests electrolyte reduction during hospitalization, although levels at discharge are similar. This is a potential confounder. Finally, we obtained data on clinical congestion that are prone to subjective evaluation, an invasive monitorization, or ultrasound evaluation, including echocardiographic, and pulmonary signs of congestion are not reported in the current analysis.

## 5. Conclusions

The management of advanced HF associated with diuretic resistance and residual congestion remains a goal for treatment, and the use of loop diuretics is still the first therapeutic option. The association of metolazone to the traditional loop diuretic therapy leads to significant weight loss and congestion signs resolution. Fluid retention and hypervolemic status resolution can be achieved by a good DR and urine output during hospitalization. Further prospective studies may be warranted to confirm our findings in a larger population.

## Figures and Tables

**Figure 1 jcm-10-04207-f001:**
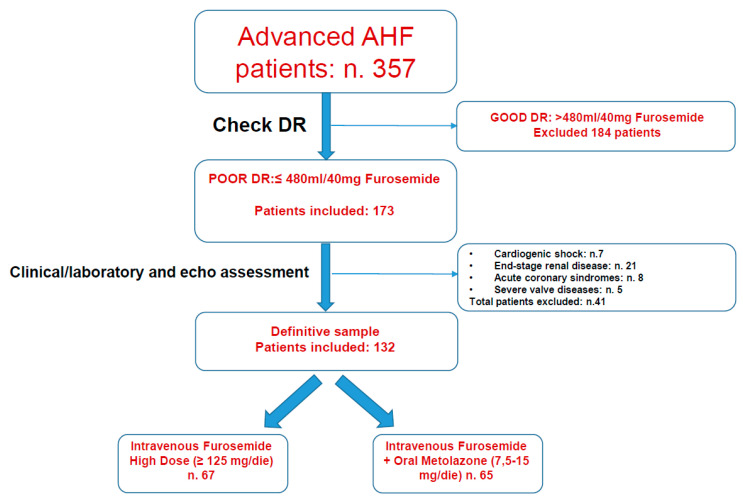
Study protocol screening diagram based on diuretic response.

**Figure 2 jcm-10-04207-f002:**
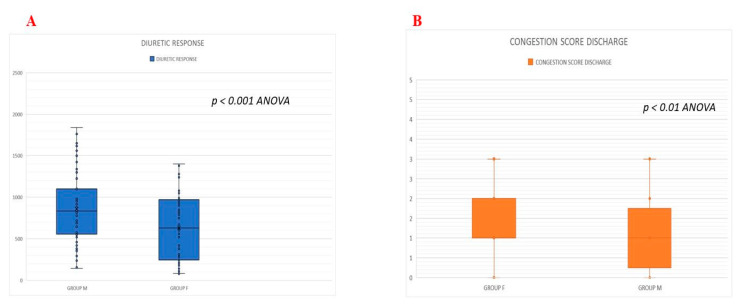
(**A**) Effects of combined metolazone plus furosemide administration (Group M) vs. furosemide alone (Group F) on diuretic response. (**B**) Clinical congestion difference assessed before discharge in Group M vs. Group F.

**Figure 3 jcm-10-04207-f003:**
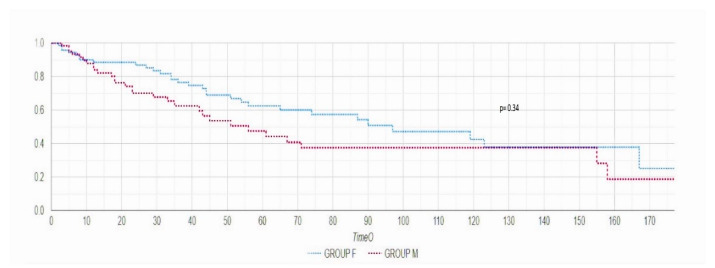
Kaplan–Meier curve during six-month follow-up period did not reveal statistical significance in terms of combined endpoint of rehospitalization and death.

**Table 1 jcm-10-04207-t001:** General characteristics of enrolled population characterized by advanced NYHA class, high congestion score, and oral diuretic administration associated with poor diuretic response.

*n* = 132 Patients	GROUP F (67 pz)	GROUP M (65 pz)
**AGE**	69 ± 16 (median 70 IQR 60–82)	71 ± 21 (median 70 IQR 58–84)
**GENDER**	32 F–38 M	17 F–46 M
**BMI (kg/m^2^)**	27 ± 7 (median 27 IQR 24–32)	28 ± 6 (median 28 IQR 24–32)
**HYPERTENSION**	55 (77%)	46 (75%)
**DYSLIPIDEMIA**	35 (49%)	28 (45%)
**DIABETES**	22 (31%)	16 (26%)
**CMD**	36 (51%)	32 (51%)
**PREVIOUS CAD**	28 (39%)	25 (40%)
**VALVULAR DISEASE**	7 (10%)	3 (9%)
**ATRIAL FIBRILLATION**	23 (32%)	21 (33%)
**MEAN EF (%)**	34 ± 6	32 ± 7
**PRO-BNP (pg/mL)**	10,316 ± 8815 (median 10,500 IQR 550–14,900)	12,177 ± 6283 (median 11,200 IQR 5600–14,300)
**CONGESTION SCORE**	3.16 ± 0.71 (median 3 IQR 2.5–3.5)	3.08 ± 0.81 (median 3 IQR 2–3.2)
**CREATININE ADMISSION (mg/dL)**	1.76 ± 1.05 (median 1.8 IQR 1.3–2.1)	1.67 ± 1.2 (median 1.6 IQR 1.3–2.1)
**eGFR (mL/min)**	39.3 ± 18 (median 38 IQR 25–45)	40.5 ± 20 (median 40 IQR 22–55)
**CKD**	35 (57%)	32 (51%)
**Blood UREA (mg/dL)**	80 ± 31 (median 78 IQR 50–75)	66 ± 17 (median 68 IQR 48–70)
**K (mEq/L)**	4.3 ± 0.5 (median 4.2 IQR 4–4.5)	4.1 ± 0.8 (median 4.1 IQR 3.6–4.4)
**Na (mEq/L)**	136 ± 5 (median 137 IQR 133–141)	137 ± 5 (median 138 IQR 136–140)
**BLOOD PRESSURE (mmHg)**	SYS 145 ± 20 DIA 81 ± 15	SYS 138 ± 18 DIA 75 ± 15
**NYHA class**	CLASS III = 17 pz (28%) CLASS IV = 51 pz (72%)	CLASS III = 15 pz (27%) CLASS IV = 50 pz (73%)
**Loop diuretic dose at admission (mg/day)**	230 ± 150 (median 225 IQR 120–300)	250 ± 120 (median 230 IQR 125–350)
**ACE-Inhibitors**	39 (58%)	36 (55%)
**Angiotensin receptor Blocker (ARB)**	20 (30%)	20 (31%)
**Beta-blockers**	41 (61%)	43 (66%)
**Digoxin**	20 (30%)	17 (26%)
**Mineralocorticoid antagonists (MRA)**	28 (42%)	26 (40%)
**Angiotensin receptor–neprilysin inhibitors (ARNIs)**	8 (12%)	9 (14%)

**Table 2 jcm-10-04207-t002:** Comparison between groups treated with furosemide alone (Group F) and furosemide plus metolazone (Group M) showing a better diuretic response, associated with significant reduction in congestion in group treated with combined diuretics.

	GROUP F	GROUP M	
**DIURETIC RESPONSE (mL/40 mg furosemide)**	541 ± 314 [median 540 IQR 940–240]	940 ± 149 [median 944 IQR 550–1080]	*p* < 0.001 ANOVA *p* < 0.03 ANCOVA
**CONGESTION SCORE** **AT DISCHARGE**	2.4 ± 1 [median 2 IQR 1.0–3.2]	1 ± 1 [median 0.8 IQR 0.5–1.2]	*p* < 0.001 ANOVA *p* = 0.03 ANCOVA
**NT-PRO-BNP (pg/mL) difference from admission to discharge**	−3954 ± 5560 [median 3900 IQR 2120–7500]	−4819 ± 8718 [median 4780 IQR 2880–12,700]	0.1
**NT-PRO-BNP (** **Δ/%** **)**	−25.1 ± 25	−26.6 ± 27.3	0.1

**Table 3 jcm-10-04207-t003:** Effects of metolazone plus furosemide (Group M) vs. furosemide alone (Group F) administration on renal function electrolyte balance and saline solution. Metolazone addition showed significant weight loss and less oral loop diuretic amount at discharge.

	GROUP F	GROUP M	
**Blood Urea (mg/dL)**	82 ± 40 [median 76 IQR 58–95]	66 ± 18 [median 67 IQR 52–75]	*p* = 0.005
**Mean eGFR (mL/min/m^2^)**	36.8 ± 22 [median 38 IQR 25–52]	38.5 ± 24 [median 39 IQR 26–50]	0.8
**CREATININE (mg/dL)**	1.69 ± 0.62 [median 1.5 IQR 1.3–2.2]	1.72 ± 0.78 [median 1.5 IQR 1.2–2.1]	0.5
**K+** **DISCHARGE (mEq/L)**	3.87 ± 0.55 [median 3.9 IQR 3.7–4.4]	4.05 ± 0.67 [median 4.1 IQR 3.6–4.2]	0.4
**NA+** **DISCHARGE (mEq/L)**	138.6 ± 4.45 [median 138 IQR 135–141]	137.8 ± 4.3 [median 138 IQR 135–141]	0.6
**HYPERTONIC SOLUTION**	8 (12%)	22 (33%)	*p* = 0.03
**MEAN DIURESIS (mL)**	2050 ± 1120 [median 2080 IQR 1650–2800]	2820 ± 900 [median 2450 IQR 1900–3000]	*p* < 0.05
**Δ WEIGHT (kg)**	−3 ± 1.5	−6 ± 2.3	*p* < 0.01
**LOOP DIURETIC DOSE** **AT DISCHARGE** **(mg)**	223.9 ± 121.7 [median 175 IQR 125–275]	175 ± 104.8 [median 150 IQR 100–250]	*p* < 0.05

## Data Availability

Not available due to patients’ privacy reasons.

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
