# Peer review of "Effects of Metolazone Administration on Congestion, Diuretic Response and Renal Function in Patients with Advanced Heart Failure"

_jcm, 2021, doi:10.3390/jcm10184207_

Round 1

Reviewer 1 Report

The authors assessed the effects of metolazone administration on congestion, diuretic response, and renal function in patients with advanced heart failure. Although the combination of diuretics with different actions is widespread, the evidence regarding this practice is surprisingly scarce.

The main limitation is that I can not see the Tables and Figures, and therefore I cannot review the manuscript.

Some comments:

- Abstract: Patients were taking 250 mg/die. I assume “die” refers to day. Are the 250 mg the furosemide dose?

- Was the dose of metalozne 7.5-15 mg per week or per day?

- Why did the authors choose such a high NTproBNP cut-off?

- Treatment with metalozone was only given during hospitalization or also at discharge?

- NTproBNP values should be given as median (interquartile range) and not mean.

- Why did the authors exclude patients with LVEF > 40?

Author Response

The authors assessed the effects of metolazone administration on congestion, diuretic response, and renal function in patients with advanced heart failure. Although the combination of diuretics with different actions is widespread, the evidence regarding this practice is surprisingly scarce.

The main limitation is that I can not see the Tables and Figures, and therefore I cannot review the manuscript.

Please accept Our sincere apologies for this trouble, we submitted all material (manuscript Tables and figures) to the journal. Perhaps the Journal system was not able to recognize the picture files . we submit again the table and figures in order you can view the whole article.

Some comments:

- Abstract: Patients were taking 250 mg/die. I assume “die” refers to day. Are the 250 mg the furosemide dose? Many thanks for this advice, we corrected accordingly

- Was the dose of metalozne 7.5-15 mg per week or per day? The metolazone  dose was  per week as reported in methods, we corrected it in the abstract

- Why did the authors choose such a high NTproBNP cut-off? Many thanks for this comment. Because contrasting findings exist regarding the beneficial/ adverse effect of metolazone, we would analyse the drug effects in patients with advanced Heart failure and poor diuretic response that in our opinion is the right target for drug administration. A hallmark feature of advanced HF in subjects with reduced EF is the increase of NTproBNP value. A lower cut off may be a potential confounder indicating less advanced HF.

- Treatment with metalozone was only given during hospitalization or also at discharge? Patients was administered even after discharge in most of cases, but our primary aim was to evaluate its effect during hospitalization and we restricted our study tom the effects during hospitalization period, For sure the post discharge administration may be a potential bias even if we did not find difference during follow up between two arms. We include this item in Limitations

- NTproBNP values should be given as median (interquartile range) and not mean. Ok we reported the value at your best preference therefore we reported in table 2  Delta NTpro BNP calculated as discharge values – admission values/admission values %, that in our opinion better reflects the temporal trend

- Why did the authors exclude patients with LVEF > 40? As previously reported, we studied patients with advanced HF and severe systolic dysfunction. We preliminary choice this cut off in order to avoid potential confounder and mix HFrEF with HFmEF.Indeed, our intention was to analyze patients with frank reduced systolic function as reported in Methods and study design

Reviewer 2 Report

The Authors presented a retrospective work based on diuretic resistance in patients advanced HF taking high loop diuretic dose. The combination of  furosemide with metolazone was evaluated retrospectively in patients with acute heart failure with reduced ejection fraction.

Overall, we believe that the present study does not add much information to the current scientific literature. As the authors themselves mention in the manuscript, the combination of loop diuretic and thiazidic diuretics has been studied previously and the greater diuretic response with sequential nephron blockade is already described. Likewise, the work presented does not have a correct design to evaluate clinical events or admissions and the results are based on congestion evaluation, despite the difficulty of evaluating it with data coming from physical examination. 

On the other hand, the study currently presents several significant limitations, some mentioned by the authors themselves and others not. Firstrly, we think that it is not consistent to reflect in the abstract or conclusions that there are no differences in renal failure or electrolytes unbalance at discharge between group M and F, if group M has received a greater number of hypertonic solutions. Secondly, within the multivariable analysis, there are a series of potential confounders that can influence diuresis, such as other diuretics or drugs with this effect.  

We kindly suggest reconsidering the study design based not so much on clinical events, but on truly objective data on diuretic response and congestion. To this end, in addition to what has been evaluated, we would recommend adding data on blood and urine hydroelectrolytes and osmolarity, as well as objective data on congestion such as the use of ultrasound to evaluate the size of the inferior vena cava, renal Doppler and lung ultrasound. The limitation of asses the congestion by pshycial examination should be mentioned as well. 

From a statistical point of view, potential biases should be ensured and minimized. It is necessary to ensure that both groups are comparable,  being a retrospective and observational selection of patients. The specific dose of furosemide in both groups should be clear. Adjusting for other drugs with diuretic effect (e.g. eplerenone, aldactone, acetazolamide, tolvaptan, iSLGT2, ARNi,...) is mandatory for a correct interpretation of the results. As mentioned, the use of the hydroelectrolyte solution should also be matched in both groups in order to correctly assess the safety results in terms of renal function and electrolytes alterations. 

Author Response

The Authors presented a retrospective work based on diuretic resistance in patients advanced HF taking high loop diuretic dose. The combination of  furosemide with metolazone was evaluated retrospectively in patients with acute heart failure with reduced ejection fraction.

Overall, we believe that the present study does not add much information to the current scientific literature. As the authors themselves mention in the manuscript, the combination of loop diuretic and thiazidic diuretics has been studied previously and the greater diuretic response with sequential nephron blockade is already described. Likewise, the work presented does not have a correct design to evaluate clinical events or admissions and the results are based on congestion evaluation, despite the difficulty of evaluating it with data coming from physical examination. Many thanks for this comment. We partially agree with the concept that metholazone was extensively examinated in AHF. Indeed the results are quite contrasting with some reports suggesting beneficial effect, whereas other studies associate drug administration with negative results. We believe that current discrepancies may depend on different patients enrolled and study target. In patients with advanced HF and reduced diuresis the combined diuretic therapy may be useful to solve congestion,to improve diuretic response and short outcome. Of note we focused our design and aim on in-hospital evaluation, then we reported followup data for completion  We added this concept in the Discussion 

On the other hand, the study currently presents several significant limitations, some mentioned by the authors themselves and others not. Firstrly, we think that it is not consistent to reflect in the abstract or conclusions that there are no differences in renal failure or electrolytes unbalance at discharge between group M and F, if group M has received a greater number of hypertonic solutions. Many thanks for this suggestion, due to the higher rate of saline administration we can hypothesize, that Group M had an higher prevalence of dyslelectrolemia despite discharge values were comparable with Group F. We inserted this sentence in Discussion. Page 8 lane 3-5

Secondly, within the multivariable analysis, there are a series of potential confounders that can influence diuresis, such as other diuretics or drugs with this effect.  We are grateful for this comments, and we performed a multivariable analysis adjusting for creatinine electrolyte NTproBNP values  and congestion score.(see new table 2) by the application of new stat test ANOVA /ANCOVA In order to avoid potential biases as explained in Exclusion criteria , we did not include patients taking other diuretics vasopressin inhibitors and other inotropic drugs

We kindly suggest reconsidering the study design based not so much on clinical events, but on truly objective data on diuretic response and congestion. To this end, in addition to what has been evaluated, we would recommend adding data on blood and urine hydroelectrolytes and osmolarity, as well as objective data on congestion such as the use of ultrasound to evaluate the size of the inferior vena cava, renal Doppler and lung ultrasound. The limitation of asses the congestion by pshycial examination should be mentioned as well. In accordance with this advice we highlighted the main results on congestion and DR. A specific paragraph describing primary and secondary endpoints has been added. As the reviewer recommended,  We added a new specific analysis matched our findings for potential confounders in order to confirm the validity.(ANCOVA test) We agree that urine electrolyte particularly Cl and Na levels, may be confirmatory respect with our findings. Unfortunately we did not have urinary electrolyte value and osmolarity results, neither congestion score evaluated by ultrasound. We inserted this item in the Limitation paragraph

From a statistical point of view, potential biases should be ensured and minimized. It is necessary to ensure that both groups are comparable,  being a retrospective and observational selection of patients. The specific dose of furosemide in both groups should be clear. Adjusting for other drugs with diuretic effect (e.g. eplerenone, aldactone, acetazolamide, tolvaptan, iSLGT2, ARNi,...) is mandatory for a correct interpretation of the results. As mentioned, the use of the hydroelectrolyte solution should also be matched in both groups in order to correctly assess the safety results in terms of renal function and electrolytes alterations. Many thanks for this suggestion: we clarified that two  groups have similar baseline characteristics and they had a similar treatment, therefore the diuretic dosage before admission was reported as well. We included a complete baseline  pharmacological treatment in Table 1 indicating the diuretic dose before admission.  Finally as explained in methods patients taking other diuretics tolvaptan or inotropic infusion were preliminarily excluded from our analysis.

The paper was revised by a native English teacher and minor language spelling mistakes have been corrected

Round 2

Reviewer 1 Report

Thank you very much for the opportunity to review the manuscript again. I have some comments:

Abstract:

- Methods: “taking a mean amount of 250 ±120 mg/day”. I assume it is furosemide, but it should be clarified. Moreover, in Table 1, the mean dose of furosemide is much lower, so I’d like to know which data is correct.
- Results: Because a change has been made, the sentence “Whereas, median NTproBNP decrease remained…” doesn’t make sense.

Methods:
Considering the metalozone treatment was given during hospitalization, it would be strange to see long-term effects. Moreover, according to the authors, some patients received metolazone after being discharged. Likely, patients in the metolazone arm were not given this treatment, and, conversely, patients in the no-M group received it at follow-up. Finally, outpatients treatment with the drug-modifying disease is more relevant prognosis-wise than the use of metolazone. I’m not sure the 6-months outcome si relevant in this study
Table:
- Please include the numeric p-value and avoid n.s. The readers would be able to interpret it.
- In English, use x.x in decimals. It is correct in some but not all numbers.
- Median values should be given with the interquartile values. The same is seen in the text.
- Table 2: Are congestion score at the admission and discharge?
Figures: please add the p-values.
Discussion:
- “Even if electrolyte vales are similar after treatment, our fundings demonstrates that Metolazone addition is associated with an increased needing of hypertonic solution.” I don’t think the authors can make this statement with the data provided. 

Author Response

Methods: “taking a mean amount of 250 ±120 mg/day”. I assume it is furosemide, but it should be clarified. Moreover, in Table 1, the mean dose of furosemide is much lower, so I’d like to know which data is correct.

Thank you for this advice the mean dose of furosemide was corrected according with text values. In the abstract we added “mean furosemide amount” and in methods was reported that we included only patients taking more than 100 mg of furosemide
- Results: Because a change has been made, the sentence “Whereas, median NTproBNP decrease remained…” doesn’t make sense. We apologize for this mistake we changed it inserting Delta Δ and  deleting decrease

Methods:
Considering the metalozone treatment was given during hospitalization, it would be strange to see long-term effects. Moreover, according to the authors, some patients received metolazone after being discharged. Likely, patients in the metolazone arm were not given this treatment, and, conversely, patients in the no-M group received it at follow-up. Finally, outpatients treatment with the drug-modifying disease is more relevant prognosis-wise than the use of metolazone. I’m not sure the 6-months outcome si relevant in this study.

We are grateful for this critical comment, Indeed the followup analysis is included among secondary endpoints. Unfortunately we have not data about treatment changes during post discharge period, and we included this statement in Limitations” Of note, some patients could modify diuretic treatment during post discharge period according to congestion presentation. Unfortunately we have only diuretic dose at discharge”
Table:
- Please include the numeric p-value and avoid n.s. The readers would be able to interpret it.
- In English, use x.x in decimals. It is correct in some but not all numbers.OK
- Median values should be given with the interquartile values. The same is seen in the text. Done,median and Interquartiles range IQR were added at baseline and at discharge in text and tables

- Table 2: Are congestion score at the admission and discharge? This is the comparison between group M and F at discharge as reported in table, we also inserted the NtproBNP difference from admission to discharge
Figures: please add the p-values. Done
Discussion:
- “Even if electrolyte vales are similar after treatment, our fundings demonstrates that Metolazone addition is associated with an increased needing of hypertonic solution.” I don’t think the authors can make this statement with the data provided. This sentence has been required by reviewer 2 ( please read the first reviewer comment). However we modified it in “metolazone treatment is associated with an increased needing of electrolyte  solution administration in order to avoid Na and K depletion during treatment”

Reviewer 2 Report

Congratulations. 

With the changes implemented by the authors, we think that the objective of the present paper is better understandable and the limitations of your work are clarified as well.   

However, there are still limitations that could hardly be solved, because they imply a profound change in the study design, initial objectives, and variables initially proposed by the authors. 

Author Response

With the changes implemented by the authors, we think that the objective of the present paper is better understandable and the limitations of your work are clarified as well.   

However, there are still limitations that could hardly be solved, because they imply a profound change in the study design, initial objectives, and variables initially proposed by the authors. 

We are grateful to the reviewer for him /her appreciation. We agree with the comment that study design and objectives are somewhat modified compared to the initial design. However, the reviewer should understand that it is a retrospective study with data collection from different centers, thus in our revision we followed the advice and we focused on the main findings requires from the reviewers. Whereas, this is the larger analysis evaluating the effects of additional Metolazone treatment in patients with advanced HF and high furosemide dose, in this sense we believe it could be additive and of relevance with respect to the current literature ; we hope the reviewer agree with this position
